# Kernel Feature Selection via Conditional Covariance Minimization

**Jianbo Chen**[*]
University of California, Berkeley
jianbochen@berkeley.edu

**Mitchell Stern**[*]
University of California, Berkeley
mitchell@berkeley.edu

**Martin J. Wainwright**
University of California, Berkeley
wainwrig@berkeley.edu

**Michael I. Jordan**
University of California, Berkeley
jordan@berkeley.edu

## Abstract

We propose a method for feature selection that employs kernel-based measures of independence to find a subset of covariates that is maximally predictive of the response. Building on past work in kernel dimension reduction, we show how to perform feature selection via a constrained optimization problem involving the trace of the conditional covariance operator. We prove various consistency results for this procedure, and also demonstrate that our method compares favorably with other state-of-the-art algorithms on a variety of synthetic and real data sets.

## 1 Introduction

Feature selection is an important issue in statistical machine learning, leading to both computational benefits (lower storage and faster computation) and statistical benefits, including increased model interpretability. With large data sets becoming ever more prevalent, feature selection has seen widespread usage across a variety of real-world tasks in recent years, including text classification, gene selection from microarray data, and face recognition [3, 14, 17]. In this work, we consider the supervised variant of feature selection, which entails finding a subset of the input features that explains the output well. This practice can reduce the computational expense of downstream learning by removing features that are redundant or noisy, while simultaneously providing insight into the data through the features that remain.

Feature selection algorithms can generally be divided into two groups: those which are agnostic to the choice of learning algorithm, and those which attempt to find features that optimize the performance of a specific learning algorithm.[1] Kernel methods have been successfully applied under each of these paradigms in recent work; for instance, see the papers [1, 8, 16, 19, 23, 25, 26, 29]. Kernel feature selection methods have the advantage of capturing nonlinear relationships between the features and the labels. Many previous approaches are filter methods based on the Hilbert-Schmidt Independence Criterion (HSIC), as proposed by Gretton et al. [13] as a measure of dependence. For instance, Song et al. [24, 25] proposed to optimize HSIC with greedy algorithms on features. Masaeli et al. [19] proposed Hilbert-Schmidt Feature Selection (HSFS), which optimizes HSIC with a continuous relaxation. In later work, Yamada et al. [29] proposed the HSIC-LASSO, in which the dual augmented Lagrangian can be used to find a global optimum. There are also wrapper methods

---

[*]Equal contribution.

[1]Feature selection algorithms that operate independently of the choice of predictor are referred to as filter methods. Algorithms tailored to specific predictors can be further divided into wrapper methods, which use learning algorithms to evaluate features based on their predictive power, and embedded methods, which combine feature selection and learning into a single problem [14].

and embedded methods using kernels. Most of the methods add weights to features and optimize the original kernelized loss function together with a penalty on the weights [1, 5, 8, 11, 12, 27, 28]. For example, Cao et al. [5] proposed margin-based algorithms for SVMs to select features in the kernel space. Lastly, Allen [1] proposed an embedded method suitable for kernel SVMs and kernel ridge regression.

In this paper, we propose to use the trace of the conditional covariance operator as a criterion for feature selection. We offer theoretical motivation for this choice and show that our method can be interpreted both as a filter method and as a wrapper method for a certain class of learning algorithms. We also show that the empirical estimate of the criterion is consistent as the sample size increases. Finally, we conclude with an empirical demonstration that our algorithm is comparable to or better than several other popular feature selection algorithms on both synthetic and real-world tasks.

## 2    Formulating feature selection

Let $\mathcal{X} \subset \mathbb{R}^d$ be the domain of covariates $X$, and let $\mathcal{Y}$ be the domain of responses $Y$. Given $n$ independent and identically distributed (i.i.d.) samples $\{(x_i, y_i),\ i = 1, 2, \ldots, n\}$ generated from an unknown joint distribution $P_{X,Y}$ together with an integer $m \leq d$, our goal is to select $m$ of the $d$ total features $X_1, X_2, \ldots, X_d$ which best predict $Y$. Let $\mathcal{S}$ be the full set of features, and let $\mathcal{T} \subseteq \mathcal{S}$ denote a subset of features. For ease of notation, we identify $\mathcal{S} = \{X_1, \ldots, X_d\}$ with $[d] = \{1, \ldots, d\}$, and also identify $X_\mathcal{T}$ with $\mathcal{T}$. We formulate the problem of supervised feature selection from two perspectives below. The first perspective motivates our algorithm as a filter method. The second perspective offers an interpretation as a wrapper method.

### 2.1    From a dependence perspective

Viewing the problem from the perspective of dependence, we would ideally like to identify a subset of features $\mathcal{T}$ of size $m$ such that the remaining features $\mathcal{S} \setminus \mathcal{T}$ are conditionally independent of the responses given $\mathcal{T}$. However, this may not be achievable when the number of allowable features $m$ is small. We therefore quantify the extent of the remaining conditional dependence using some metric $\mathcal{Q}$, and aim to minimize $\mathcal{Q}$ over all subsets $\mathcal{T}$ of the appropriate size. More formally, let $\mathcal{Q} : 2^{[d]} \to [0, \infty)$ be a function mapping subsets of $[d]$ to the non-negative reals that satisfies the following properties:

- For a subset of features $\mathcal{T}$, we have $\mathcal{Q}(\mathcal{T}) = 0$ if and only if $X_{\mathcal{S} \setminus \mathcal{T}}$ and $Y$ are conditionally independent given $X_\mathcal{T}$.
- The function $\mathcal{Q}$ is non-increasing, meaning that $\mathcal{Q}(\mathcal{T}) \geq \mathcal{Q}(\mathcal{S})$ whenever $\mathcal{T} \subseteq \mathcal{S}$. Hence, the function $\mathcal{Q}$ achieves its minimum for the full feature set $\mathcal{T} = [d]$.

Given a fixed integer $m$, the problem of supervised feature selection can then be posed as

$$\min_{\mathcal{T} : |\mathcal{T}| = m} \mathcal{Q}(\mathcal{T}). \tag{1}$$

This formulation can be taken as a filter method for feature selection.

### 2.2    From a prediction perspective

An alternative perspective aims at characterizing how well $X_\mathcal{T}$ can predict $Y$ directly within the context of a specific learning problem. Formally, we define the error of prediction as

$$\mathcal{E}_\mathcal{F}(X) = \inf_{f \in \mathcal{F}} \mathbb{E}_{X,Y} L(Y, f(X)), \tag{2}$$

where $\mathcal{F}$ is a class of functions from $\mathcal{X}$ to $\mathcal{Y}$, and $L$ is a *loss function* specified by the user. For example, in a univariate regression problem, the function class $\mathcal{F}$ might be the set of all linear functions, and the loss function might be the squared error $L(Y, f(X)) = (Y - f(X))^2$.

We then hope to solve the following problem:

$$\min_{\mathcal{T} : |\mathcal{T}| \leq m} \mathcal{E}_\mathcal{F}(X_\mathcal{T}) = \min_{\mathcal{T} : |\mathcal{T}| \leq m} \inf_{f \in \mathcal{F}_m} \mathbb{E}_{X,Y} L(Y, f(X_\mathcal{T})),$$

where $\mathcal{F}_m$ is a class of functions supported on $\mathbb{R}^m$. That is, we aim to find the subset of $m$ features that minimizes the prediction error. This formulation thus falls within the scope of wrapper methods for feature selection.

## 3 Conditional covariance operator

The conditional covariance operator provides a measure of conditional dependence for random variables. It was first proposed by Baker [2], and was further studied and used for sufficient dimension reduction by Fukumizu et al. [9, 10]. We provide a brief overview of this operator and some of its key properties here.

Let $(\mathbb{H}_{\mathcal{X}}, k_{\mathcal{X}})$ and $(\mathbb{H}_{\mathcal{Y}}, k_{\mathcal{Y}})$ denote reproducing kernel Hilbert spaces (RKHSs) of functions on spaces $\mathcal{X}$ and $\mathcal{Y}$, respectively. Also let $(X, Y)$ be a random vector on $\mathcal{X} \times \mathcal{Y}$ with joint distribution $P_{X,Y}$. Assume the kernels $k_{\mathcal{X}}$ and $k_{\mathcal{Y}}$ are bounded in expectation:

$$\mathbb{E}_X[k_{\mathcal{X}}(X, X)] < \infty \quad \text{and} \quad \mathbb{E}_Y[k_{\mathcal{Y}}(Y, Y)] < \infty. \tag{3}$$

The cross-covariance operator associated with the pair $(X, Y)$ is the mapping $\Sigma_{YX} : \mathbb{H}_{\mathcal{X}} \to \mathbb{H}_{\mathcal{Y}}$ defined by the relations

$$\langle g, \Sigma_{YX} f \rangle_{\mathbb{H}_{\mathcal{Y}}} = \mathbb{E}_{X,Y}[(f(X) - \mathbb{E}_X[f(X)])(g(Y) - \mathbb{E}_Y[g(Y)])] \quad \text{for all } f \in \mathbb{H}_X \text{ and } g \in \mathbb{H}_Y. \tag{4}$$

Baker [2] showed there exists a unique bounded operator $V_{YX}$ such that

$$\Sigma_{YX} = \Sigma_{YY}^{1/2} V_{YX} \Sigma_{XX}^{1/2}. \tag{5a}$$

The conditional covariance operator is then defined as

$$\Sigma_{YY|X} = \Sigma_{YY} - \Sigma_{YY}^{1/2} V_{YX} V_{XY} \Sigma_{YY}^{1/2}. \tag{5b}$$

Among other results, Fukumizu et al. [9, 10] showed that the conditional covariance operator captures the conditional variance of $Y$ given $X$. More precisely, if the sum $\mathbb{H}_{\mathcal{X}} + \mathbb{R}$ is dense in $L^2(P_X)$, where $L^2(P_X)$ is the space of all square-integrable functions on $\mathcal{X}$, then we have

$$\langle g, \Sigma_{YY|X} g \rangle_{\mathbb{H}_{\mathcal{Y}}} = \mathbb{E}_X[\text{var}_{Y|X}[g(Y)|X]] \quad \text{for any } g \in \mathbb{H}_{\mathcal{Y}}. \tag{6}$$

From Proposition 2 in the paper [10], we also know the residual error of $g(Y)$ with $g \in \mathbb{H}_{\mathcal{Y}}$ can be characterized by the conditional covariance operator. More formally, for any $g \in \mathbb{H}_{\mathcal{Y}}$, we have

$$\langle g, \Sigma_{YY|X} g \rangle_{\mathbb{H}_{\mathcal{Y}}} = \inf_{f \in \mathcal{H}_{\mathcal{X}}} \mathbb{E}_{X,Y}((g(Y) - \mathbb{E}_Y[g(Y)]) - (f(X) - \mathbb{E}_X[f(X)]))^2. \tag{7}$$

## 4 Proposed method

In this section, we describe our method for feature selection, which we call conditional covariance minimization (CCM).

Let $(H_1, k_1)$ denote an RKHS supported on $\mathcal{X} \subset \mathbb{R}^d$. Let $\mathcal{T} \subseteq [d]$ be a subset of features with cardinality $m \leq d$, and for all $x \in \mathbb{R}^d$, take $x^{\mathcal{T}} \in \mathbb{R}^d$ to be the vector with components $x_i^{\mathcal{T}} = x_i$ if $i \in \mathcal{T}$ or 0 otherwise. We define the kernel $k_1^{\mathcal{T}}$ by $k_1^{\mathcal{T}}(x, \tilde{x}) = k_1(x^{\mathcal{T}}, \tilde{x}^{\mathcal{T}})$ for all $x, \tilde{x} \in \mathcal{X}$. Suppose further that the kernel $k_1$ is permutation-invariant. That is, for any $x, \tilde{x} \in \mathcal{X}$ and permutation $\pi$, denoting $(x_{\pi(1)}, \dots, x_{\pi(d)})$ as $x_\pi$, we have $k_1(x, \tilde{x}) = k_1(x_\pi, \tilde{x}_\pi)$. (Note that this property holds for many common kernels, including the linear, polynomial, Gaussian, and Laplacian kernels.) Then for every $\mathcal{T}$ of cardinality $m$, $k_1^{\mathcal{T}}$ generates the same RKHS supported on $\mathbb{R}^m$. We call this RKHS $(\tilde{H}_1, \tilde{k}_1)$. We will show the trace of the conditional covariance operator $\text{trace}(\Sigma_{YY|X})$ can be interpreted as a dependence measure, as long as the RKHS $H_1$ is large enough.

We say that an RKHS $(H, k)$ is characteristic if the map $P \to \mathbb{E}_P[k(X, \cdot)] \in H$ is one-to-one. If $k$ is bounded, this is equivalent to saying that $H + \mathbb{R}$ is dense in $L^2(P)$ for any probability measure $P$ [10]. We have the following lemma, whose proof is given in the appendix:

**Lemma 1.** *If $k_1$ is bounded and characteristic, then $\tilde{k}_1$ is also characteristic.*

Let $(H_2, k_2)$ denote an RKHS supported on $\mathcal{Y}$. Based on the above lemma, we have the following theorem, which is a parallel version of Theorem 4 in [10]:

**Theorem 2.** *If $(H_1, k_1)$ and $(H_2, k_2)$ are characteristic, we have $\Sigma_{YY|X} \preceq \Sigma_{YY|X_{\mathcal{T}}}$ with equality holding if and only if $Y \perp\!\!\!\perp X | X_{\mathcal{T}}$.*

The proof is postponed to the appendix.

With this generic result in place, we now narrow our focus to problems with univariate responses, including univariate regression, binary classification and multi-class classification. In the case of regression, we assume $H_2$ is supported on $\mathbb{R}$, and we take $k_2$ to be the linear kernel:

$$k_2(y, \tilde{y}) = y\tilde{y} \tag{8}$$

for all $y, \tilde{y} \in \mathbb{R}$. For binary or multi-class classification, we take $k_2$ to be the Kronecker delta function:

$$k_2(y, \tilde{y}) = \delta(y, \tilde{y}) = \begin{cases} 1 & \text{if } y = \tilde{y}, \\ 0 & \text{otherwise.} \end{cases} \tag{9}$$

This can be equivalently interpreted as a linear kernel $k(y, \tilde{y}) = \langle y, \tilde{y} \rangle$ assuming a one-hot encoding of $Y$, namely that $\mathcal{Y} = \{y \in \{0,1\}^k : \sum_i y_i = 1\} \subset \mathbb{R}^k$, where $k$ is the number of classes.

When $\mathcal{Y}$ is $\mathbb{R}$ or $\{y \in \{0,1\}^k : \sum_i y_i = 1\} \subset \mathbb{R}^k$, we obtain the following corollary of Theorem 2:

**Corollary 3.** *If $(H_1, k_1)$ is characteristic, $\mathcal{Y}$ is $\mathbb{R}$ or $\{y \in \{0,1\}^k : \sum_i y_i = 1\} \subset \mathbb{R}^k$, and $(H_2, k_2)$ includes the identity function on $\mathcal{Y}$, then we have $\mathrm{Tr}(\Sigma_{YY|X}) \leq \mathrm{Tr}(\Sigma_{YY|X_{\mathcal{T}}})$ for any subset $\mathcal{T}$ of features. Moreover, the equality $\mathrm{Tr}(\Sigma_{YY|X}) = \mathrm{Tr}(\Sigma_{YY|X_{\mathcal{T}}})$ holds if and only if $Y \perp\!\!\!\perp X|X_{\mathcal{T}}$.*

Hence, in the univariate case, the problem of supervised feature selection reduces to minimizing the trace of the conditional covariance operator over subsets of features with controlled cardinality:

$$\min_{\mathcal{T}:|\mathcal{T}|=m} \mathcal{Q}(\mathcal{T}) := \mathrm{Tr}(\Sigma_{YY|X_{\mathcal{T}}}). \tag{10}$$

In the regression setting, Equation (7) implies the residual error of regression can also be characterized by the trace of the conditional covariance operator when using the linear kernel on $\mathcal{Y}$. More formally, we have the following observation:

**Corollary 4.** *Let $\Sigma_{YY|X_{\mathcal{T}}}$ denote the conditional covariance operator of $(X_{\mathcal{T}}, Y)$ in $(\tilde{H}_1, \tilde{k}_1)$. Define the space of functions $\mathcal{F}_m$ from $\mathbb{R}^m$ to $\mathcal{Y}$ as*

$$\mathcal{F}_m = \tilde{H}_1 + \mathbb{R} := \{f + c : f \in \tilde{H}_1, c \in \mathbb{R}\}. \tag{11}$$

*Then we have*

$$\mathrm{Tr}(\Sigma_{YY|X_{\mathcal{T}}}) = \mathcal{E}_{\mathcal{F}_m}(X_{\mathcal{T}}) = \inf_{f \in \mathcal{F}_m} \mathbb{E}_{X,Y}(Y - f(X_{\mathcal{T}}))^2. \tag{12}$$

Given the fact that the trace of the conditional covariance operator can characterize the dependence and the prediction error in regression, we will use the empirical estimate of it as our objective. Given $n$ samples $\{(x_1, y_1), \ldots, (x_n, y_n)\}$, the empirical estimate is given by [10]:

$$\mathrm{trace}(\hat{\Sigma}_{YY|X_{\mathcal{T}}}^{(n)}) := \mathrm{trace}[\hat{\Sigma}_{YY}^{(n)} - \hat{\Sigma}_{YX_{\mathcal{T}}}^{(n)}(\hat{\Sigma}_{X_{\mathcal{T}}X_{\mathcal{T}}}^{(n)} + \varepsilon_n I)^{-1}\hat{\Sigma}_{X_{\mathcal{T}}Y}^{(n)}]$$

$$= \varepsilon_n \, \mathrm{trace}[G_Y(G_{X_{\mathcal{T}}} + n\varepsilon_n I_n)^{-1}],$$

where $\hat{\Sigma}_{YX}^{\mathcal{T}(n)}, \hat{\Sigma}_{X_{\mathcal{T}}X}^{\mathcal{T}(n)}$ and $\hat{\Sigma}_{YY}^{(n)}$ are the covariance operators defined with respect to the empirical distribution and $G_{X_{\mathcal{T}}}$ and $G_Y$ are the centralized kernel matrices, respectively. Concretely, we define

$$G_{X_{\mathcal{T}}} := (I_n - \frac{1}{n}\mathbb{1}\mathbb{1}^T)K_{X_{\mathcal{T}}}(I_n - \frac{1}{n}\mathbb{1}\mathbb{1}^T) \quad \text{and} \quad G_Y := (I_n - \frac{1}{n}\mathbb{1}\mathbb{1}^T)K_Y(I_n - \frac{1}{n}\mathbb{1}\mathbb{1}^T).$$

The $(i, j)$th entry of the kernel matrix $K_{X_{\mathcal{T}}}$ is $\tilde{k}_1(x_{\mathcal{T}}^i, x_{\mathcal{T}}^j)$, with $x_{\mathcal{T}}^i$ denoting the $i$th sample with only features in $\mathcal{T}$. As the kernel $k_2$ on the space of responses is linear, we have $K_Y = \mathbf{Y}\mathbf{Y}^T$, where $\mathbf{Y}$ is the $n \times k$ matrix with each row being a sample response. Without loss of generality, we assume each column of $\mathbf{Y}$ is zero-mean, so that $G_Y = K_Y = \mathbf{Y}\mathbf{Y}^T$. Our objective then becomes:

$$\mathrm{trace}[G_Y(G_{X_{\mathcal{T}}} + n\varepsilon_n I_n)^{-1}] = \mathrm{trace}[\mathbf{Y}\mathbf{Y}^T(G_{X_{\mathcal{T}}} + n\varepsilon_n I_n)^{-1}] = \mathrm{trace}[\mathbf{Y}^T(G_{X_{\mathcal{T}}} + n\varepsilon_n I_n)^{-1}\mathbf{Y}]. \tag{13}$$

For simplicity, we only consider univariate regression and binary classification where $k = 1$, but our discussion carries over to the multi-class setting with minimal modification. The objective becomes

$$\min_{|\mathcal{T}|=m} \hat{\mathcal{Q}}^{(n)}(\mathcal{T}) := \mathbf{y}^T(G_{X_{\mathcal{T}}} + n\varepsilon_n I_n)^{-1}\mathbf{y}, \tag{14}$$

where $\mathbf{y} = (y_1, \ldots, y_n)^T$ is an $n$-dimensional vector. We show the global optimal of the problem (14) is consistent. More formally, we have the following theorem:

**Theorem 5** (Feature Selection Consistency). *Let the set $A = argmin_{|\mathcal{T}| \leq m} \mathcal{Q}(\mathcal{T})$ be the set of all the optimal solutions to (12) and $\hat{T}^{(n)} \in argmin_{|T| \leq m} \hat{\mathcal{Q}}^{(n)}(\mathcal{T})$ be a global optimal of (14). If $\varepsilon_n \to 0$ and $\varepsilon_n n \to \infty$ as $n \to \infty$, we have*

$$P(\hat{T}^{(n)} \in A) \to 1. \tag{15}$$

Our proof is provided in the appendix. A comparable result is given in Fukumizu et al. [10] for the consistency of their dimension reduction estimator, but as our minimization takes place over a finite set, our proof is considerably simpler.

## 5  Optimization

Finding a global optimum for (14) is NP-hard for generic kernels [28], and exhaustive search is computationally intractable if the number of features is large. We therefore approximate the problem of interest via continuous relaxation, as has previously been done in past work on feature selection [4, 27, 28].

### 5.1  Initial relaxation

We begin by introducing a binary vector $w \in \{0,1\}^d$ to indicate which features are active. This allows us to rephrase the optimization problem from (14) as

$$
\begin{aligned}
\min_{w} \quad & \mathbf{y}^T (G_{w \odot X} + n\varepsilon_n I_n)^{-1} \mathbf{y} \\
\text{subject to} \quad & w_i \in \{0,1\}, \; i = 1, \ldots, d, \\
& \mathbb{1}^T w = m,
\end{aligned} \tag{16}
$$

where $\odot$ denotes the Hadamard product between two vectors and $G_{w \odot X}$ is the centralized version of the kernel matrix $K_{w \odot X}$ with $(K_{w \odot X})_{ij} = k_1(w \odot x_i, w \odot x_j)$.

We then approximate the problem (16) by relaxing the domain of $w$ to the unit hypercube $[0,1]^d$ and replacing the equality constraint with an inequality constraint:

$$
\begin{aligned}
\min_{w} \quad & \mathbf{y}^T (G_{w \odot X} + n\varepsilon_n I_n)^{-1} \mathbf{y} \\
\text{subject to} \quad & 0 \leq w_i \leq 1, \; i = 1, \ldots, d, \\
& \mathbb{1}^T w \leq m.
\end{aligned} \tag{17}
$$

This objective can be optimized using projected gradient descent, and represents our first tractable approximation. A solution to the relaxed problem is converted back into a solution for the original problem by setting the $m$ largest values of $w$ to 1 and remaining values to 0. We initialize $w$ to the uniform vector $(m/d)[1, 1, \ldots, 1]^T$ in order to avoid the corners of the constraint set during the early stages of optimization.

### 5.2  Computational issues

The optimization problem can be approximated and manipulated in a number of ways so as to reduce computational complexity. We discuss a few such options below.

**Removing the inequality constraint.**  The hard constraint $\mathbb{1}^T w \leq m$ requires a nontrivial projection step, such as the one detailed in Duchi et al. [6]. We can instead replace it with a soft constraint and move it to the objective. Letting $\lambda_1 \geq 0$ be a hyperparameter, this gives rise to the modified problem

$$
\begin{aligned}
\min_{w} \quad & \mathbf{y}^T (G_{w \odot X} + n\varepsilon_n I_n)^{-1} \mathbf{y} + \lambda_1 (\mathbb{1}^T w - m) \\
\text{subject to} \quad & 0 \leq w_i \leq 1, \; i = 1, \ldots, d.
\end{aligned} \tag{18}
$$

**Removing the matrix inverse.** The matrix inverse in the objective function is an expensive operation. In light of this, we first define an auxiliary variable $\alpha \in \mathbb{R}^n$, add the equality constraint $\alpha = (G_{w \odot X} + n\varepsilon_n I_n)^{-1}\mathbf{y}$, and rewrite the objective as $\alpha^T \mathbf{y}$. We then note that we may multiply both sides of the constraint by the centered kernel matrix to obtain the relation $(G_{w \odot X} + n\varepsilon_n I_n)\alpha = \mathbf{y}$. Letting $\lambda_2 \geq 0$ be a hyperparameter, we finally replace this relation by a soft $\ell_2$ constraint to obtain

$$
\begin{aligned}
\min_{w,\alpha} \quad & \alpha^T \mathbf{y} + \lambda_2 \|(G_{w \odot X} + n\varepsilon_n I_n)\alpha - \mathbf{y}\|_2^2 \\
\text{subject to} \quad & 0 \leq w_i \leq 1, \ i = 1, \dots, d, \\
& \mathbb{1}^T w \leq m.
\end{aligned}
\tag{19}
$$

**Using a kernel approximation.** Rahimi and Recht [22] propose a method for approximating kernel evaluations by inner products of random feature vectors, so that $k(x, \tilde{x}) \approx z(x)^T z(\tilde{x})$ for a random map $z$ depending on the choice of kernel $k$. Let $K_w \approx U_w U_w^T$ be such a decomposition, where $U_w \in \mathbb{R}^{n \times D}$ for some $D < n$. Then, defining $V_w = (I - \mathbb{1}\mathbb{1}^T/n)U_w$, we similarly have that the centered kernel matrix can be written as $G_w \approx V_w V_w^T$. By the Woodbury matrix identity, we may write

$$
\begin{aligned}
(G_{w \odot X} + n\varepsilon_n I_n)^{-1} &\approx \frac{1}{\varepsilon_n n} I - \frac{1}{\varepsilon_n^2 n^2} V_w (I_D + \frac{1}{\varepsilon_n n} V_w^T V_w)^{-1} V_w^T \\
&= \frac{1}{\varepsilon_n n}(I - V_w(V_w^T V_w + \varepsilon_n n I_D)^{-1} V_w^T).
\end{aligned}
\tag{20}
$$

Substituting this into our objective function, scaling by $\epsilon_n n$, and removing the constant term $\mathbf{y}^T \mathbf{y}$ resulting from the identity matrix gives a new approximate optimization problem. This modification reduces the complexity of each optimization step from $\mathcal{O}(n^2 d + n^3)$ to $\mathcal{O}(n^2 D + D^3 + nDd)$.

**Choice of formulation.** We remark that each of the three approximations beyond the initial relaxation may be independently used or omitted, allowing for a number of possible objectives and constraint sets. We explore some of these configurations in the experimental section below.

# 6 Experiments

In this section, we evaluate our approach (CCM) on both synthetic and real-world data sets. We compare with several strong existing algorithms, including recursive feature elimination (RFE) [15], Minimum Redundancy Maximum Relevance (mRMR) [21], BAHSIC [24, 25], and filter methods using mutual information (MI) and Pearson's correlation (PC). We use the author's implementation for BAHSIC[2] and use Scikit-learn [20] and Scikit-feature [17] packages for the rest of the algorithms. The code for our approach is publicly available at `https://github.com/Jianbo-Lab/CCM`.

## 6.1 Synthetic data

We begin with experiments on the following synthetic data sets:

- Binary classification (Friedman et al. [7]). Given $Y = -1$, $(X_1, \dots, X_{10}) \sim N(0, I_{10})$. Given $Y = 1$, $X_1$ through $X_4$ are standard normal conditioned on $9 \leq \sum_{j=1}^4 X_j^2 \leq 16$, and $(X_5, \dots, X_{10}) \sim N(0, I_6)$.

- 3-dimensional XOR as 4-way classification. Consider the 8 corners of the 3-dimensional hypercube $(v_1, v_2, v_3) \in \{-1, 1\}^3$, and group them by the tuples $(v_1 v_3, v_2 v_3)$, leaving 4 sets of vectors paired with their negations $\{v^{(i)}, -v^{(i)}\}$. Given a class $i$, a point is generated from the mixture distribution $(1/2)N(v^{(i)}, 0.5I_3) + (1/2)N(-v^{(i)}, 0.5I_3)$. Each example additionally has 7 standard normal noise features for a total of 10 dimensions.

- Additive nonlinear regression: $Y = -2\sin(2X_1) + \max(X_2, 0) + X_3 + \exp(-X_4) + \varepsilon$, where $(X_1, \dots, X_{10}) \sim N(0, I_{10})$ and $\varepsilon \sim N(0, 1)$.

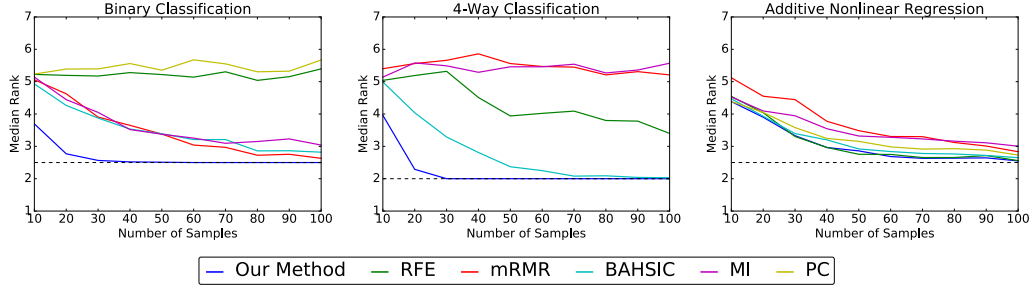

Figure 1: The above plots show the median rank ($y$-axis) of the true features as a function of sample size ($x$-axis) for the simulated data sets. Lower median ranks are better. The dotted line indicates the optimal median rank.

The first data set represents a standard nonlinear binary classification task. The second data set is a multi-class classification task where each feature is independent of $Y$ by itself but a combination of three features has a joint effect on $Y$. The third data set arises from an additive model for nonlinear regression.

Each data set has $d = 10$ dimensions in total, but only $m = 3$ or $4$ true features. Since the identity of these features is known, we can evaluate the performance of a feature selection algorithm by computing the median rank it assigns to the real features, with lower median ranks indicating better performance. Given enough samples, we would expect this value to come close to the optimal lower bound of $(m + 1)/2$.

Our experimental setup is as follows. We generate 10 independent copies of each data set with sample sizes ranging from 10 to 100, and record the median ranks assigned to the true features by each algorithm. This process is repeated a total of 100 times, and the results are averaged across trials. For kernel-based methods, we use a Gaussian kernel $k(x, \tilde{x}) = \exp(-\|x - \tilde{x}\|^2/(2\sigma^2))$ on $X$ and a linear kernel $k(y, \tilde{y}) = y^T \tilde{y}$ on $Y$. We take $\sigma$ to be the median pairwise distance between samples scaled by $1/\sqrt{2}$. Since the number of true features is known, we provide this as an input to algorithms that require it.

Our initial experiments use the basic version of our algorithm from Section 5.1. When the number of desired features $m$ is fixed, only the regularization parameter $\varepsilon$ needs to be chosen. We use $\varepsilon = 0.001$ for the classification tasks and $\varepsilon = 0.1$ for the regression task, selecting these values from $\{0.001, 0.01, 0.1\}$ using cross-validation. Our results are shown in Figure 1.

On the binary and 4-way classification tasks, our method outperforms all other algorithms, succeeding in identifying the true features using fewer than 50 samples where others require close to 100 or even fail to converge. On the additive nonlinear model, several algorithms perform well, and our method is on par with the best of them across all sample sizes.

These experiments show that our algorithm is comparable to or better than several widely-used feature selection techniques on a selection of synthetic tasks, and is adept at capturing several kinds of nonlinear relationships between the covariates and the responses. When compared in particular to its closest relative BAHSIC, a backward-elimination algorithm based on the Hilbert–Schmidt independence criterion, we see that our algorithm often produces higher quality results with fewer samples, and even succeeds in the non-additive problem where BAHSIC fails to converge.

We also rerun these experiments separately for each of the first two approximations described in Section 5.2 above, selecting $\lambda_1$ from $\{0.001, 0.01, 0.1\}$ and $\lambda_2$ from $\{1, 10, 100\}$ using cross-validation. We find that comparable results can be attained with either approximate objective, but note that the algorithm is more robust to changes in $\lambda_1$ than $\lambda_2$.

## 6.2 Real-world data

In the previous section, we found that our method for feature selection excelled in identifying nonlinear relationships on a variety of synthetic data sets. We now turn our attention to a collection

|          | ALLAML | CLL-SUB-111 | glass | ORL | orlraws10P | pixraw10P | TOX-171 | vowel | warpAR10P | warpPIE10P | wine | Yale |
|----------|--------|-------------|-------|-----|------------|-----------|---------|-------|-----------|------------|------|------|
| Samples  | 72     | 111         | 214   | 400 | 100        | 100       | 171     | 990   | 130       | 210        | 178  | 165  |
| Features | 7,129  | 11,340      | 10    | 1,024 | 10,304   | 10,000    | 5,784   | 10    | 2,400     | 2,420      | 13   | 1,024 |
| Classes  | 2      | 3           | 6     | 40  | 10         | 10        | 4       | 11    | 10        | 10         | 3    | 15   |

Table 1: Summary of the benchmark data sets we use for our empirical evaluation.

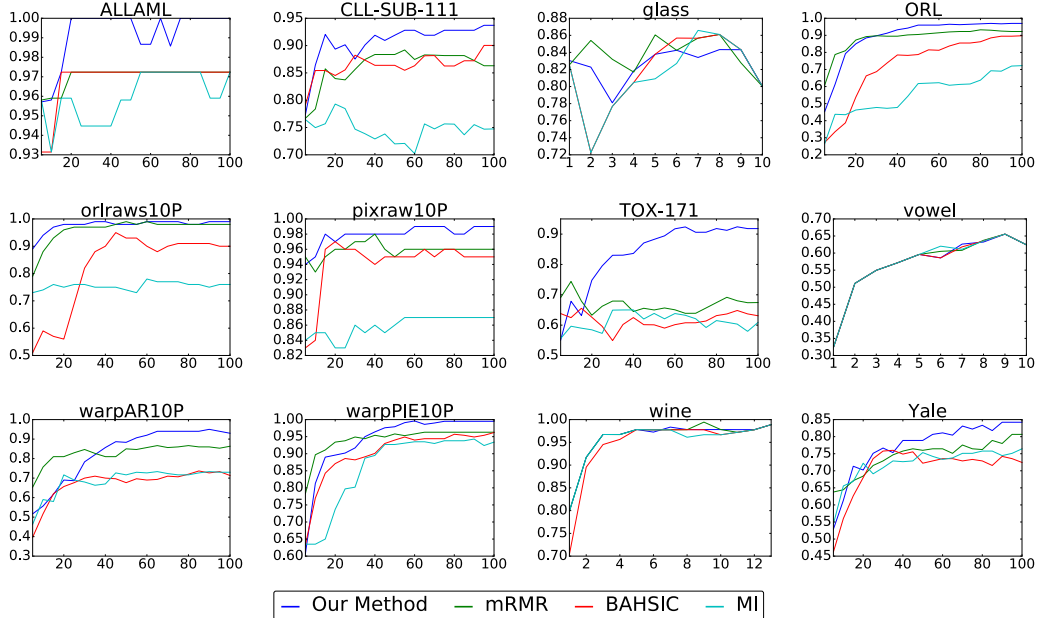

Figure 2: The above plots show classification accuracy ($y$-axis) versus number of selected features ($x$-axis) for our real-world benchmark data sets. Higher accuracies are better.

of real-word tasks, studying the performance of our method and other nonlinear approaches when used in conjunction with a kernel SVM for downstream classification.

We carry out experiments on 12 standard benchmark tasks from the ASU feature selection website [17] and the UCI repository [18]. A summary of our data sets is provided in Table 1. The data sets are drawn from several domains including gene data, image data, and voice data, and span both the low-dimensional and high-dimensional regimes.

For every task, we run each algorithm being evaluated to obtain ranks for all features. Performance is then measured by training a kernel SVM on the top $m$ features and computing the resulting accuracy as measured by 5-fold cross-validation. This is done for $m \in \{5, 10, \ldots, 100\}$ if the total number of features $d$ is larger than 100, or $m \in \{1, 2, \ldots, d\}$ otherwise. In all cases we fix the regularization constant of the SVM to $C = 1$ and use a Gaussian kernel with $\sigma$ set as in the previous section over the selected features. For our own algorithm, we fix $\varepsilon = 0.001$ across all experiments and set the number of desired features to $m = 100$ if $d > 100$ or $\lceil d/5 \rceil$ otherwise. Our results are shown in Figure 2.

Compared with three other popular methods for nonlinear feature selection, i.e. mRMR, BAHSIC, and MI, we find that our method is the strongest performer in the large majority of cases, sometimes by a substantial margin as in the case of TOX-171. While our method is occasionally outperformed in the beginning when the number of selected features is small, it either ties or overtakes the leading method by the end in all but one instance. We remark that our method consistently improves upon the performance of the related BAHSIC method, suggesting that our objective based on conditional covariance may be more powerful than one based on the Hilbert-Schmidt independence criterion.

# 7 Conclusion

In this paper, we proposed an approach to feature selection based on minimizing the trace of the conditional covariance operator. The idea is to select the features that maximally account for the dependence of the response on the covariates. We do so by relaxing from an intractable discrete formulation of the problem to a continuous approximation suitable for gradient-based optimization. We demonstrate the effectiveness of our approach on multiple synthetic and real-world experiments, finding that it often outperforms other state-of-the-art approaches, including another competitive kernel feature selection method based on the Hilbert-Schmidt independence criterion.

## Footnotes

[2] `http://www.cc.gatech.edu/~lsong/code.html`

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
