[Supplementary Material]

# A Appendix

The appendix is devoted to proofs of various results from the main text.

## A.1 Proof of Lemma 1

Suppose there exist two distributions $P, Q$ on $\mathbb{R}^m$ such that $\mathbb{E}_{P(X)}[\widetilde{k}(y, X)] = \mathbb{E}_{Q(X)}[\widetilde{k}_1(y, X)]$ for any $y \in \mathbb{R}^m$. Consider $\mathbb{R}^m$ as a subspace embedded in $\mathbb{R}^d$. The probability distributions $P$ and $Q$ can be extended to $\mathbb{R}^d$ by by setting the remaining components to zero. Then we also have $\mathbb{E}_{P(X)}[\widetilde{k}_1(y, X)] = \mathbb{E}_{Q(X)}[\widetilde{k}_1(y, X)]$ for any $y \in \mathbb{R}^d$. As $k_1$ is characteristic, $P = Q$.

## A.2 Proof of Theorem 2 and Corollary 3

Theorem 2 and Corollary 3 can be proved simultaneously. The proof of Theorem 2 parallels the proof of the corresponding theorem in the setting of dimension dimension reduction by Fukumizu et al. [10].

We can interpret $\widetilde{\mathbb{H}}_1$ as a subset of $\mathbb{H}_1$, so Equation 7 implies $\Sigma_{YY|X_{\mathcal{T}}} \geq \Sigma_{YY|X}$. In the univariate case, this is equivalent to saying $\text{trace}[\Sigma_{YY|X_{\mathcal{T}}}] \geq \text{trace}[\Sigma_{YY|X}]$.

By the law of total variance, we have for any $g \in \mathbb{H}_2$,

$$\mathbb{E}_{X_{\mathcal{T}}} \text{var}_{Y|X_{\mathcal{T}}}[g(Y)|X_{\mathcal{T}}] = \mathbb{E}_X[\text{var}[g(Y)|X]] + \mathbb{E}_{X_{\mathcal{T}}} \text{var}_{Y|X_{\mathcal{T}}}[\mathbb{E}_{Y|X}[g(Y)|X]]. \quad (21)$$

By Lemma 1, the kernel $\widetilde{k}_1$ is characteristic, so the conditional covariance operator characterizes the conditional dependence, which reduces Equation 21 to

$$\langle g, (\Sigma_{YY|X_{\mathcal{T}}} - \Sigma_{YY|X})g \rangle = \mathbb{E}_{X_{\mathcal{T}}} \text{var}_{Y|X_{\mathcal{T}}}[\mathbb{E}_{Y|X}[g(Y)|X]]. \quad (22)$$

Hence $\Sigma_{YY|X_{\mathcal{T}}} = \Sigma_{YY|X}$ if and only if given $X_{\mathcal{T}}$, $\mathbb{E}_{Y|X}[g(Y)|X]$ is almost surely determined. Because $k_2$ is characteristic, we have $Y \perp\!\!\!\perp X|X_{\mathcal{T}}$. Suppose $Y$ is univariate and $k_2$ is the linear kernel. Then both $\Sigma_{YY|X}$ and $\Sigma_{YY|X_{\mathcal{T}}}$ can be equivalently interpreted as linear functions that map real numbers to real numbers. When $g = \text{Id}_Y$, the identity function on $Y$, we have

$$\langle \text{Id}_Y, (\Sigma_{YY|X_{\mathcal{T}}} - \Sigma_{YY|X})\text{Id}_Y \rangle = \mathbb{E}_{X_{\mathcal{T}}} \text{var}_{Y|X_{\mathcal{T}}}[\mathbb{E}_{Y|X}[Y|X]] = 0. \quad (23)$$

This implies $Y \perp\!\!\!\perp X|X_{\mathcal{T}}$ directly.

## A.3 Proof of Theorem 5

We provide a simpler proof than the one for Theorem 6 in the paper [10], where the consistency result for dimension reduction was established.

For any subset of features $\mathcal{T}$, we have

$$|\text{trace}[\hat{\Sigma}_{YY|X_{\mathcal{T}}}] - \text{trace}[\Sigma_{YY|X_{\mathcal{T}}}]|$$
$$\leq |\text{trace}[\Sigma_{YY|X_{\mathcal{T}}}] - \text{trace}[\Sigma_{YY} - \Sigma_{YX_{\mathcal{T}}}(\Sigma_{X_{\mathcal{T}}X_{\mathcal{T}}} + \varepsilon_n I)^{-1}\Sigma_{X_{\mathcal{T}}Y}]|+$$
$$+ |\text{trace}[\Sigma_{YY} - \Sigma_{YX_{\mathcal{T}}}(\Sigma_{X_{\mathcal{T}}X_{\mathcal{T}}} + \varepsilon_n I)^{-1}\Sigma_{X_{\mathcal{T}}Y}] - \text{trace}[\hat{\Sigma}_{YY|X_{\mathcal{T}}}]|,$$

where the second term converges to zero by the law of large numbers, whereas Fukumizu et al. [10] proved that the second term can be upper bounded as

$$\frac{1}{\varepsilon_n}\{(\|\hat{\Sigma}_{YX_{\mathcal{T}}^{(n)}}\|_{\text{HS}} + \|\Sigma_{YX_{\mathcal{T}}}\|_{HS})\|\hat{\Sigma}_{YX_{\mathcal{T}}} - \Sigma_{YX_{\mathcal{T}}}\|_{\text{HS}}$$
$$+ \|\Sigma_{YY}\|_{\text{trace}}\|\hat{\Sigma}_{X_{\mathcal{T}}X_{\mathcal{T}}}^{(n)} - \Sigma_{X_{\mathcal{T}}X_{\mathcal{T}}}\|_{\text{HS}}$$
$$+ |\text{trace}[\hat{\Sigma}_{YY} - \Sigma_{YY}]|\},$$

where $\|\cdot\|_{\text{HS}}$ is the HSIC norm of an operator. By the Central Limit Theorem, both of the terms

$$\|\hat{\Sigma}_{YX_{\mathcal{T}}} - \Sigma_{YX_{\mathcal{T}}}\|_{\text{HS}}, \|\hat{\Sigma}_{X_{\mathcal{T}}X_{\mathcal{T}}}^{(n)} - \Sigma_{X_{\mathcal{T}}X_{\mathcal{T}}}\|_{\text{HS}} \quad \text{and} \quad |\text{trace}[\hat{\Sigma}_{YY} - \Sigma_{YY}]|$$

are guaranteed to be of order $\mathcal{O}_p(n^{-1/2})$. Hence, the second term also converges to 0. This establishes the convergence of $\text{trace}[\hat{\Sigma}_{YY|X_{\mathcal{T}}}]$ towards $\text{trace}[\Sigma_{YY|X_{\mathcal{T}}}]$, which yields the claim (15) by standard $\varepsilon$–$\delta$ arguments.