[Reviews · NeurIPS 2017]

Reviewer 1



In this paper, authors propose a new nonlinear feature selection based on kernels. More specifically, the conditional covariance operator has been employed to measure the conditional independence between Y and X given the subset of X. Then, the feature selection can be done by searching a set of features that minimizing the conditional independence. This optimization problem results in minimizing over matrix inverse and it is hard to optimize it. Thus, a novel approach to deal with the matrix inverse problem is also proposed. Finally, the consistency of the feature selection algorithm is proved. Through experiments, the authors showed that the proposed algorithm compares favorably with existing algorithms. The paper is clearly written and the proposed algorithm is technically sound. Overall, I like the algorithm. Detailed comments: 1. In the proposed algorithm, some information about tuning parameters are missing. So, please include those information in the paper. For example, lambda_2 in Eq. (20). Also, how to initialize the original w? 2. It would be great to compare with other kernel based feature selection algorithms such as HSIC-LASSO and KNIFE. Also, related to the first comments, is it possible to further improve the proposed algorithm by properly initializing the w? For instance, HSIC-LASSO is a convex approach and can have reasonable features. 3. The results of TOX-171 is pretty interesting. It is nice to further investigate the reason why the proposed approach outperforms rest of it. 4. It is nice to have comparison of redundancy rate. In feature selection community tends to use this measure for evaluation. Zhao, Z., Wang, L., and Li, H. (2010). Efficient spectral feature se lection with minimum redundancy. In AAAI Conference on Artificial Intelligence (AAAI) , pages 673–678.

Reviewer 2



In this paper, authors propose a feature selection method that minimizes the conditional covariance of the output Y in RKHS given the selected features. It can be understood as minimizing the "scatterness" of Y given X. Clearly, a good feature selection should minimize such a variance. Several relaxations are also introduced to mitigate the computational cost. The methodology is sound and the paper is well-written. It may contain an interesting contribution to the field of kernel feature selection. I have only a few concerns: 1. According to the definition of conditional independence, PY|X = PY|X_T. Can't we just minimize the difference between PY|X and PY|X_T? If the mapping from the distribution to the mean embedding is one-to-one, can't we just minimize the distance between these two conditional mean embeddings, using conditional MMD? What is the advantage of using a covariance operator or considering the second order moment? 2. The computation of the proposed method looks heavy (see line 189) even after using the kernel approximation. How does it compare with the HSIC based methods? The performance of different relaxations should also have been tested in experiments. In general, this paper contains an interesting solution to kernel feature selection problems, though the computational advantage versus HSIC based methods is still not very clear to me.

Reviewer 3



Feature selection based on conditional covariance operator in RKHS proposed. The basic idea of the proposed method is in line with kernel dimension reduction (reference [9] in the manuscript), and the approach for optimization is simple (put weights for each feature dimension and minimize the proposed trace criterion of conditional covariance.) Namely, the concept and methodology and not very new(but well thought-out), but the most important contribution I think of this paper is proving the feature selection consistency (Theorem 2). I feel it is a significant contribution and this paper is worth accepted. In "Gradient-based kernel method for feature extraction and variable selection,", NIPS2012, a kernel-based feature selection method is also presented. The approach is different but the one in NIPS2012 and the one in the current submission both are based on conditional covariance operator in RKHS. It would be informative to mention the work in NIPS2012 and give some comments on the similarity and differences. Minor issues: - line 94: The notation "[d]" should be "\mathcal{S}", as it is introduced in line 48. - line 131,132: F_m should be \mathcal{F}_m, and I couldn't find H_1^k in eq.(12). - line 146: "Q(T)" should be "\mathcal{Q}(\mathcal{T})", and |T|\leq m in line 147 should be |\mathcal{T}| \leq m. - line 147: Isn't it better to refer to eq.(15) instead of eq.(17)? - line 173: \|w_1\| \leq m - The epsilon parameter is set to 0.001 and 0.1 for classification and regression tasks, resp., but I'd like to know the sensitivity of the proposed methods on the parameter.